# Mef2c- and Nkx2-5-Divergent Transcriptional Regulation of Chick WT1_76127 and Mouse Gm14014 lncRNAs and Their Implication in Epicardial Cell Migration

**DOI:** 10.3390/ijms252312904

**Published:** 2024-11-30

**Authors:** Sheila Caño-Carrillo, Carlos Garcia-Padilla, Amelia E. Aranega, Estefania Lozano-Velasco, Diego Franco

**Affiliations:** 1Cardiovascular Development Group, Department of Experimental Biology, University of Jaen, 23071 Jaen, Spain; scano@ujaen.es (S.C.-C.); cgpadill@ujaen.es (C.G.-P.); aaranega@ujaen.es (A.E.A.); evelasco@ujaen.es (E.L.-V.); 2Fundación Medina, 18016 Granada, Spain

**Keywords:** lncRNAs, epicardial cell, cytoskeletal remodeling, cell migration

## Abstract

Cardiac development is a complex developmental process. The early cardiac straight tube is composed of an external myocardial layer and an internal endocardial lining. Soon after rightward looping, the embryonic heart becomes externally covered by a new epithelial lining, the embryonic epicardium. A subset of these embryonic epicardial cells migrate and colonize the embryonic myocardium, contributing to the formation of distinct cell types. In recent years, our understanding of the molecular mechanisms that govern proepicardium and embryonic epicardium formation has greatly increased. We have recently witnessed the discovery of a novel layer of complexity governing gene regulation with the discovery of non-coding RNAs. Our laboratory recently identified three distinct lncRNAs, adjacent to the *Wt1*, *Bmp4* and *Fgf8* chicken gene loci, with enhanced expression in the proepicardium that are distinctly regulated by Bmp, Fgf and thymosin β4, providing support for their plausible implication in epicardial formation. The expression of lncRNAs was analyzed in different chicken and mouse tissues as well as their subcellular distribution in chicken proepicardial, epicardial, ventricle explants and in different murine cardiac cell types. lncRNA transcriptional regulation was analyzed by using siRNAs and expression vectors of different transcription factors in chicken and mouse models, whereas antisense oligonucleotides were used to inhibit *Gm14014* expression. Furthermore, RT-qPCR, immunocytochemistry, RNA pulldown, Western blot, viability and cell migration assays were conducted to investigate the biological functions of *Wt1_76127* and *Gm14014*. We demonstrated that *Wt1_76127* in chicken and its putative conserved homologue *Gm14014* in mice are widely distributed in different embryonic and adult tissues and distinctly regulated by cardiac-enriched transcription factors, particularly *Mef2c* and *Nkx2.5*. Furthermore, silencing assays demonstrated that mouse *Gm14014*, but not chicken *Wt1_76127*, is essential for epicardial, but not endocardial or myocardial, cell migration. Such processes are governed by partnering with Myl9, promoting cytoskeletal remodeling. Our data show that *Gm14014* plays a pivotal role in epicardial cell migration essential for heart regeneration under these experimental conditions.

## 1. Introduction

Cardiac development is a complex developmental process that initiates soon after gastrulation with the configuration of symmetrical pools of cardiomyogenic precursors that subsequently fuse in the embryonic midline, leading to the formation of the cardiac straight tube [1]. This early tube is composed of an external myocardial layer and an internal endocardial lining. Shortly after, rightward looping develops and the embryonic atrial and ventricular chambers progressively develop [2]. At this stage, the embryonic heart becomes externally covered by a new epithelial lining, the embryonic epicardium (EE). The EE originates from the proepicardium (PE), a transient cauliflower-like structure located at the junction of the cardiac and hepatic anlagen within the septum transversum [3]. Cells emanating from the PE bridge to the naked embryonic myocardium providing an external lining [4]. Soon after, a subset of these embryonic epicardial cells undergoes an epithelial-to-mesenchymal transition (EMT), colonizing the subepicardial space and migrating into the embryonic myocardium [5,6]. In chicken, these epicardial-derived cells (i.e., EPDCs) have been consistently reported to contribute to distinct coronary vascular components, including the vascular endothelium, smooth muscle and adventitial layers, as well as cardiac fibroblasts [7,8]. However, in mice, several lineage-tracing experiments have provided evidence of a modest contribution of EPDCs to the cardiac endothelium, while their contribution to coronary smooth muscle and adventitial vasculature and the fibroskeleton is undisputed [9].

In recent years, our understanding of the molecular mechanisms that govern proepicardial and epicardial formation has greatly emerged [3]. Evidence highlighting the pivotal role of Bmp and Fgf during proepicardial and myocardial specification was reported by Kruithof et al. [10] demonstrating that Fgf enhances proepicardial specification while Bmp promotes cardiomyogenic differentiation. More recently, the role of distinct transcription factors has emerged as key players in epicardial development. *Wt1* is essential for epicardial EMT and maturation [11,12,13,14,15,16]. Similarly, *Tbx18* has also been reported to be expressed in the embryonic epicardium but its role in epicardial formation remains controversial. Greulich et al. [17] reported that *Tbx18* is dispensable for epicardial formation, EMT and subsequent differentiation into smooth muscle cells and fibroblasts while Wu et al. [18] demonstrated impaired coronary plexus formation and identified the impaired expression of several signaling pathways related to vascular development, such as Hedgehog, Vegf, Angiopoetin and Wnt signaling. Further support for the role of *Tbx18* in epicardial EMT was reported by Takeichi et al. [19] using murine primary epicardial cells. *Epicardin/Tcf21* is also discretely expressed in the PE and EE [20,21] and regulates the specification and maturation of the proepicardial cells [22] in *Xenopus*, while in mice it has been reported to be essential for epicardial-derived fibroblast formation [23] and the inhibition of smooth muscle differentiation [24]. Other cardiac transcription factors with pivotal roles in cardiogenesis, including *Gata4*, *Nkx2.5*, *Isl1* and *Pitx2* have been reported during PE/EE formation [25,26], yet their functional contributions are still uncertain, except that of *Gata4* which is essential for PE formation [27].

For decades, the adult epicardium was considered merely an external cardiac lining with no or limited physiological implications. However, it has become clearly demonstrated that upon injury, the epicardium is reactivated, providing essential cues for regenerating the damaged heart in experimental models such as zebrafish, medaka and mice [28,29,30,31,32,33]. The direct contribution of epicardial cells is limited [34]; however, adult thymosin β4-primed epicardial cells can be converted into fully functional integrated cardiomyocytes after myocardial infarction [35,36,37], supporting the therapeutic potential of the epicardium in healing the damaged heart [38,39,40,41]. Furthermore, EPDCs play determinant roles in distinct cardiac pathological conditions, contributing to an increased fibrous response after myocardial infarction [42] as well as providing cellular substrates for atrial fat deposition [43] that correlates with an enhanced prevalence of arrhythmogenic diseases such as atrial fibrillation [44,45,46].

Over the last decade, we have witnessed the rise of a novel layer of complexity governing gene regulation with the discovery of non-coding RNAs [47,48]. Non-coding RNAs are broadly classified according to their length into small non-coding RNAs (<200 nt) and long non-coding RNAs (>200 nt) [47,48]. microRNAs constitute the most abundantly expressed and widely studied types of RNAs among small non-coding RNAs. microRNAs are molecules that are 20–24 nucleotides in length that play essential roles in controlling the post-transcriptional regulation of coding RNAs through base–pair complementary binding within the 3′UTRs and promoting mRNA degradation and/or protein translation blockage [47]. On the other hand, long non-coding RNAs undergo a similar biogenesis process to coding RNAs, but despite their length, no protein coding potential has been identified [48]. Several lncRNAs have been reported to exert pivotal roles during early embryonic development, such as *Fendrr* and *Braveheart* [49,50], but none of them have been identified during epicardial formation. Furthermore, some lncRNAs have been identified to be distinctly expressed in cardiac pathological conditions such as myocardial ischemia, heart failure or arrhythmogenic diseases [51,52,53,54].

Recently, we identified three distinct lncRNAs, adjacent to the *Wt1*, *Bmp4* and *Fgf8* chicken gene loci, with enhanced expression in the PE compared to the embryonic myocardium in the developing embryonic chicken heart. These lncRNAs are distinctly regulated by Bmp, Fgf and thymosin β4, providing support for their plausible implication in epicardial cell lineage specification [55]. Therefore, within this study, we provide a thorough characterization of their tissue and subcellular distribution, their transcriptional regulation, their conservation in mice and their functional role. Herein, we provide evidence that these lncRNAs are widely distributed and transcriptionally regulated by cardiac enriched-transcription factors. Furthermore, mouse *Gm14014*, but not its chicken homologue *Wt1_76127*, is required for epicardial cell migration.

## 2. Results

### 2.1. Tissue Distribution in the Embryonic Chicken

Gene expression and tissue distribution analyses are important for understanding the plausible functional roles of lncRNAs as they provide hints about their cellular functions [56]. Therefore, we analyzed the expression profiles of *Wt1_76127*, *Fgf8_57126* and *Bmp4_53170* lncRNAs in the distinct embryonic structures of HH24 and HH32 chicken embryos using RT-qPCR. Our data demonstrate that *Wt1_76127* and *Fgf8_57126* are predominantly expressed in limb buds and eyes with residual expression in the heart at HH24 (Figure 1A). On the other hand, *Bmp4_53170* is preferentially expressed in the eye and upper limb buds (Figure 1A). At HH32, the expression of *Wt1_76127* remains highest in the limb buds followed by the body wall, head, eye and liver with minimal expression in the heart, similarly to its expression at HH24 (Figure 1B). *Fgf8_57126* displays a rather similar expression profile to *Wt1_76127* (Figure 1B), while *Bmp4_53170* displays the highest expression in the body wall and liver with moderate expression in the heart and limb buds (Figure 1B). Overall, these data demonstrate that *Wt1_76127* and *Fgf8_57126* display similar expression patterns during embryogenesis, in contrast to *Bmp4_53170*. Nevertheless, all three of them are widely distributed across different tissues during embryogenesis. These data therefore suggest that these lncRNAs might be functionally relevant in multiple tissues.

### 2.2. Subcellular Distribution in Proepicardial, Epicardial and Ventricular Tissues During Development

Cardiovascular diseases are the most prominent cause of death worldwide [57,58,59]. The impairment of distinct molecular signaling pathways greatly contributes to such a large incidence. In this context, lncRNAs have been extensively reported to be distinctly located in different subcellular compartments, exerting epigenetic and transcriptional roles if they are preferentially distributed in the nucleus and post-transcriptional roles if they are distributed in the cytoplasm. We are, therefore, particularly interested in understanding the functional role of these lncRNAs in the developing and adult heart. Therefore, we analyzed their subcellular distribution in the cytoplasmic and nuclear compartments of proepicardial (HH17), epicardial (HH24 and HH32) and ventricular (HH17, HH24 and HH32) tissues (Figure 1C). Our data demonstrate that *Wt1_76127*, *Fgf8_57126* and *Bmp4_53170* display a prominent nuclear localization in proepicardial and epicardial cells within all of the stages analyzed (Figure 1D). On the other hand, while the expression of *Wt1_76127*, *Fgf8_57126* and *Bmp4_53170* is prominently nuclear during the early developmental ventricular stages, i.e., HH17, they also become progressively expressed in the cytoplasm at HH24 and HH32, with *Fgf8_57126* and *Bmp4_53170* being almost equally represented, while *Wt1_76127* continues to be predominantly nuclear (Figure 1D). Thus, these data support the notion that these lncRNAs might preferentially exert epigenetic and transcriptional roles.

### 2.3. Transcriptional Regulation of Wt1_76127, Fgf8_57126 and Bmp4_53170

Several studies demonstrate the ability of lncRNAs to regulate the expression of transcription factors [60]. However, the expression of lncRNAs is also a transcriptionally regulated process [61], although this is scarcely investigated. We sought to investigate whether different cardiac-enriched transcription factors, i.e., *Mef2c*, *Nkx2.5*, *Pitx2c* and *Srf*, can regulate the expression of the *Wt1_76127*, *Fgf8_57126* and *Bmp4_53170* lncRNAs in epicardial and ventricular explants. To investigate this, we performed loss-of-function assays in HH24 epicardial and ventricular explants and transfected them with their corresponding siRNAs (loss-of-function) as previously reported [62] (Appendix A). Silencing *Mef2c* in epicardial explants significantly diminishes the expression of *Bmp4_53170* and *Fgf8_57126,* while *Wt1_76127* is not altered (Figure 1E). *Nkx2.5* siRNA administration leads to an up-regulation of *Wt1_76127* and *Fgf8_57126.* while *Bmp4_53170* expression is down-regulated. On the other hand, silencing *Pitx2c* and *Srf* leads to a down-regulation of all three lncRNAs (*Wt1_76127*, *Bmp4_53170* and *Fgf8_57126*), except for *Fgf8_57126* that is not significantly altered after *Srf* siRNA administration (Figure 1E). Overall, these data demonstrate that *Mef2c*, *Pitx2c* and *Srf* are essential for the transcriptional activation of these three lncRNAs while *Nkx2.5* acts as a repressor for *Wt1_76127* and *Fgf8_57126* expression in epicardial explants.

Analyses of the results of the loss-of-function assays in the ventricular explants demonstrate the distinct transcriptional regulation carried out by these transcription factors. Silencing *Mef2c* leads to up-regulation of all three lncRNAs, i.e., *Wt1_76127*, *Fgf8_57126* and *Bmp4_53170*, in HH24 ventricular explants (Figure 1F). *Nkx2.5* silencing only decreases *Bmp4_53170,* while *Wt1_76127* and *Fgf8_57126* are not altered. *Pitx2c* silencing specifically diminishes *Wt1_76127* without affecting *Bmp4_53170* or *Fgf8_57126*. Similarly, *Srf* silencing in ventricular explants only decreases *Bmp4_53170* and *Wt1_76127* but *Fgf8_57126* is not modified (Figure 1F). Thus, these data demonstrate that *Mef2c* plays a fundamental role repressing the expression of all three lncRNAs in the ventricular explants while *Nkx2.5, Pitx2* and *Srf* have a rather limited role, only barely modulating *Wt1_76127* and *Bmp4_53170* expression in ventricular explants. Interestingly, our data also demonstrate that the regulation of these lncRNAs is distinctly modulated by different cardiac-enriched transcription factors in epicardial vs. ventricular explants.

Since we have previously observed that these lncRNAs display a dynamic subcellular distribution during embryonic development, we sought to investigate if silencing these transcription factors might differentially regulate their subcellular distribution in the embryonic epicardium, focusing particularly on *Mef2c*. The loss of the function of *Mef2c* in epicardial explants in HH24 did not modify the nuclear vs. cytoplasmic distribution of *Wt1_76127*, enhanced *Bmp4_53170* expression in both subcellular compartments and selectively down-regulated *Fgf8_57126* expression in the nucleus while not altering cytoplasmic expression (Appendix A). The nuclear vs. cytoplasmic distribution of *Fgf8_57126* in HH32 remained consistent as that in HH24 after the loss of the function of *Mef2c*, although it negatively regulated *Wt1_76127’s* nuclear distribution and *Bmp4_53170’s* cytoplasmic distribution, respectively (Appendix A). Thus, these data demonstrate that cytoplasmic vs. nuclear localization is modulated in part by *Mef2c*.

### 2.4. Wt1_76127 Is Conserved in the Mouse (Gm14014)

The conservation of lncRNAs across different species is rarely identified, as lncRNAs seem to be less conserved than coding RNAs [48,62]. We sought to investigate if *Wt1_76127*, *Fgf8_57126* and *Bmp4_53170* are conserved in the mouse genome. An analysis of the annotated lncRNAs in the mouse genome led to the identification of *Gm14014* as a putative homologue of *Wt1_76127* since it is located in the vicinity of the Wt1 locus and it shares a 39,1% homology identity with *Wt1_76127* (Appendix A). Comparative analyses of such nucleotide conservation display a wide and heterogeneous distribution of such conserved nucleotide stretches in *Gm14014* (Appendix A). On the other hand, no homologues for *Fgf8_57126* and *Bmp4_53170* were found.

### 2.5. Gm14014 Is Widely Expressed in Mouse Embryonic Tissues

To investigate whether the homology between *Wt1_76127* and *Gm14014* correlates with similar expression patterns, we tested *Gm14014* expression in multiple tissues during embryonic development. RT-qPCR analyses demonstrated that, in mouse E13.5 embryos, *Gm14014* is expressed at higher levels in the lungs, stomach, kidney and liver compared to the head (Figure 2A). Analyses of *Gm14014* by RT-qPCR in adult mice also displayed a wide tissue distribution (Figure 2B). Overall, these data are in line with those observed for *Wt1_76127* in chicken embryos, characterized by widespread tissue distribution and low expression levels in the developing heart.

To gain further insights into the tissue distribution of *Gm14014*, single-molecule SCRINSHOT analyses were performed in different embryonic (E10.5, E13.5, E16.5, E19.5) and postnatal (P2, P7, P21) hearts. The *Gm14014* SCRINSHOT analyses showed a similar tissue distribution in both the embryonic and postnatal stages (Figure 2C,D), indicating the specific location of *Gm14014*-positive cells (Appendix A). The number of *Gm14014*-positive cells was analyzed during each developmental stage, displaying the highest expression at E13.5 (9.5% of the total cell counts). From this stage onwards, the *Gm14014* levels progressively decreased, with the lowest expression level at P21 (0.32%), except for postnatal stage P2 (5.98%), when the expression peaked at similar levels to those in the earliest embryonic stages (Figure 2E). Subsequently, colocalization analyses with muscle (*Tnnt2*) [63], epicardial (*Col1a2*) [64] and endocardial (*Pecam*) [65] markers were performed (Figure 2F,G). The highest *Gm14014* colocalization was identified with *Pecam* and *Tnnt2* at E13.5 (10.04% and 10.29%, respectively) (Figure 2G). In line with the global expression results, *Gm14014*-positive cells displayed the lowest colocalization levels at postnatal stages P7 and P21, while at P2 they were similar to those during the early embryonic stages (Figure 2G). Therefore, these data demonstrate that *Gm14014* is widely distributed in different cell types in both embryonic and adult hearts, displaying higher expression levels during the embryonic stages.

### 2.6. Gm14014 Is Nuclearly Located and Displays Tissue-Specific Transcriptional Regulation

To further characterize the cardiovascular expression of *Gm14014* and thus determine its plausible biological role in this context, we analyzed the expression of *Gm14014* in different cardiac cell lines, as well as its subcellular localization. Moreover, since the regulation of lncRNA expression is similar to that of coding RNAs, we investigated the impact of growth factors and key transcription factors involved in cardiogenesis on *Gm14014* expression. Our data demonstrate that the highest expression of *Gm14014* is observed in endocardial MEVEC cells, followed by EPIC and MEC1 epicardial cells, with the lowest expression in HL1 atrial cardiomyocytes (Figure 3A). Importantly, the subcellular localization analyses carried out using RT-qPCR showed that *Gm14014* is predominantly located in the nucleus across all four distinct cell lines (Figure 3B and Appendix A).

Several members of the Fgf and Bmp families, e.g., *Bmp2*, *Bmp4*, *Fgf2* and *Fgf8*, are essential players for cell fate determination during cardiogenesis, particularly during PE/EE development [10,66]. We recently demonstrated that *Wt1_76127*, *Fgf8_57126* and *Bmp4_53170* are distinctly regulated by these growth factors as well as by thymosin β4. We investigated whether these growth factors could also modulate *Gm14104* expression in different cardiac cell lines. Within epicardial (MEC1) cells, *Gm14014* expression is increased after *Fgf2* and *Bmp4* administration, whereas the addition of *Bmp2* displayed no significant effect and *Fgf8* and thymosin β4 significantly down-regulated it (Figure 3C). Additionally, in endocardial MEVEC cells, *Fgf2*, *Fgf8* and thymosin β4 administration resulted in significant *Gm14014* down-regulation while *Bmp2* and *Bmp4* led to no significant differences (Figure 3C) while in HL1 cells, *Fgf8*, *Bmp2* and thymosin β4 administration significantly down-regulate *Gm14014* expression, whereas *Fgf2* and *Bmp4* resulted in no significant differences (Figure 3C). Therefore, these data demonstrate that both Fgfs and Bmps can distinctly modulate *Gm14014* in different cardiac cell types. Curiously, thymosin β4 consistently down-regulated *Gm14014* expression in all three cardiovascular cell types.

We subsequently tested the functional role of distinct cardiac-enriched transcription factors in these four different cell lines using gain- and loss-of-function assays for the *Mef2c*, *Nkx2.5*, *Pitx2c* and *Srf* transcription factors (Appendix A). Within HL1 atrial cardiomyocytes, *Gm14014* is significantly down-regulated by *Nkx2.5* and *Srf* inhibition, while an over-expression of these transcription factors did not significantly modulate its expression (Figure 3D,E). On the other hand, *Pitx2c* over-expression, but not inhibition, significantly up-regulated *Gm14014* (Figure 3D,E). Overall, these data demonstrate that *Mef2c*, *Nkx2.5* and *Srf* are indispensable for *Gm1401*4 expression whereas *Pitx2c* on its own is capable of enhancing *Gm14014* expression in cardiomyocytes.

*Gm14104* expression in MEC1 epicardial cells is equally up-regulated by the gain and the loss of the function of *Mef2c*, while *Nkx2.5* inhibition significantly down-regulates and *Nkx2.5* over-expression up-regulates *Gm14014* (Figure 3D,E). In contrast, *Srf* inhibition significantly up-regulates *Gm14014*, while *Srf* over-expression down-regulates it, thus showing opposite regulatory capabilities to *Nkx2.5* (Figure 3D,E). On the other hand, only *Pitx2c* inhibition selectively down-regulates *Gm14014* while *Pitx2c* over-expression does not modulate its expression (Figure 3D,E). Therefore, these data demonstrate that *Nkx2.5* and *Srf* play fundamental and opposite roles in regulating *Gm14014* in MEC1 epicardial cells, while only *Pitx2c* silencing modulates its expression.

Within EPIC epicardial cells, the modulatory roles of these transcription factors seem to be substantially different to those for MEC1 epicardial cells, likely correlating with the epithelial nature of MEC1 as compared to EPIC, which exhibits a mixed epithelial and mesenchymal behavior. *Mef2c* and *Nkx2.5* silencing does not modulate *Gm14014* whereas *Pitx2c* and *Srf* inhibition significantly up-regulates its expression (Figure 3D,E). On the other hand, *Mef2c* over-expression does not modulate *Gm14014* while *Nkx2.5*, *Pitx2c* and *Srf* significantly down-regulates *Gm14014, respectively* (Figure 3D,E). Therefore, *Pitx2c* and *Srf* are essential factors for *Gm14014* expression in EPIC cells and only *Srf* displays similar regulatory effects in MEC1 and EPIC cells.

Finally, within MEVEC endocardial cells, both *Mef2c* inhibition and over-expression significantly down-regulated *Gm14014* expression (Figure 3D,E), while *Nkx2.5* and *Pitx2c* inhibition significantly down-regulated *Gm14014,* with no significant alterations in its expression observed upon its over-expression (Figure 3D,E). On the other hand, *Srf* inhibition down-regulated while *Srf* over-expression up-regulated *Gm14014* in MEVEC endocardial cells (Figure 3D,E). Therefore, all four transcription factors analyzed are essential for *Gm14014* expression in MEVEC endocardial cells, yet only *Srf* over-expression is capable of inducing *Gm14014* expression. Consequently, these data demonstrate that the expression of *Gm14014* is regulated in part by *Mef2c*, *Nkx2.5* and *Srf* in HL1 cardiomyocytes, *Nkx2.5* and *Pitx2c* in MEC1 epicardial cells and *Mef2c*, *Nkx2.5*, *Pitx2c* and *Srf* in MEVEC endocardial cells. However, no critical dependence on any of the tested transcription factors was observed in EPIC cells, as revealed by the siRNA assays. Importantly, these data further support the notion that distinct cardiac-enriched transcription factors modulate *Gm14014* expression in different cell types, in line with similar observations made for *Wt1_76127* transcriptional regulation in epicardial and ventricular explants (Figure 1E,F). In this context it is important to highlight that *Mef2c* exerts a similar regulatory role in *Wt1_76127* and *Gm14014* in epicardial explants and MEC1 cells, supporting the pivotal role of this transcription factor in this context.

### 2.7. Identification of Gm14014 Interacting Proteins

LncRNAs can exert their functions by interacting with different types of molecules, including other RNAs species and proteins. Importantly, dissecting lncRNA–protein interactions can provide hints into their functional roles [67,68]. To gain further insights into the molecular mechanisms driven by the lncRNA *Gm14014* in an epicardial context, we performed RNA pull-down (PD) assays followed by mass spectrometry (MS) to identify *Gm14014*-associated proteins in epicardial MEC1 cells. A total of 315 proteins were uniquely identified in the *Gm14014* PD assays as compared to input and *Gapdh* PD controls (Appendix A); in total, ~25% were exclusively nuclear, ~30% were exclusively cytoplasmic and ~28% were present in both compartments (Appendix A). Gene ontology (GO) analyses of uniquely identified proteins in *Gm141014* PD revealed their primary involvement in biological processes such as mRNA processing, translation, mRNA processing and RNA splicing, protein transport, the positive regulation of gene expression, cytoplasmic translation, RNA splicing via the spliceosome and actin cytoskeleton organization (Appendix A). These results highlight the prominent role of *Gm14014* in association with nuclear biological processes, in line with its predominant nuclear localization, although it is important to note that distinct cytoplasmic biological functions were also identified.

GO analyses of cellular components revealed cytoplasm, cytosol, nucleus, nucleoplasm, cytoskeleton synapse and nucleolus as the most representative compartments (Appendix A). Finally, a GO molecular function analysis identified protein binding, nucleotide binding, RNA binding, identical protein binding, nucleic acid binding, ATP binding and actin binding as the most represented functions (Appendix A). Thus, all of the GO analyses supported the dual nuclear and cytoplasmic role for *Gm14014*.

We subsequently focused our attention on those cellular compartments and molecular and biological functions with a higher number of uniquely identified proteins in *Gm14014* PD, i.e., in the cytoplasm, protein binding and translation. Among these categories, several families of proteins were highly represented, such as ribosylation factors, DEAD box helicases, small GTPases RAB proteins, splicing factors, ribosomal proteins and cytoskeletal proteins. Additionally, multiple cytoskeletal proteins were identified as interacting with *Gm14014*, including those controlling actin filament organization (Rac1), positive and negative actin polymerization (cofilin, profilin, gelsolin, filamin) and actin-interacting proteins (actins, actinins, myosins, talin, tubulin, tropomyosin and vinculin). These findings suggest, therefore, that *Gm14014* might be involved in cytoskeletal organization, potentially contributing to cytoskeletal remodeling and thus to cell migration.

### 2.8. Dissecting the Functional Role of Gm14014 in Epicardial Cell Migration

As mentioned above, epicardial cell migration is a critical process during embryonic development, enabling the external coverage of the myocardium as well as promoting epicardial-derived cells to invade the myocardium. Importantly, epicardial cell migration is not only required during embryogenesis, but also after cardiac injury when epicardial cells are activated to promote cardiac regeneration. To elucidate a plausible role for *Gm14014* in the epicardial cell migration process, we performed a loss-of-function assay on *Gm14014* in an MEC1 epicardial cell line. Two distinct ASOs were designed and it was demonstrated that ASO1, but not ASO2, effectively inhibited *Gm14014* expression (Appendix A) after 6 h of transfection. Thereafter, we tested the timeframe of *Gm14014* ASO1 inhibition, demonstrating that consistent down-regulation was also observed at 12 h and 18 h, while at 24 h inhibition was blunted and an overt up-regulation was promoted (Figure 4A). Surprisingly, 48h after transfection, a down-regulation was detected again (Figure 4A). Based on these results, all *Gm14014* inhibitions were carried out only with ASO1 (i.e., ASO from now on).

To determine the effect of *Gm14014* inhibition on the migration process, we carried out scratch cell migration assays on the semi-confluent MEC1 epicardial cells. Analyses of the migratory cells demonstrated that *Gm14014* ASO-treated cells were consistently delayed at 8, 12 and 24 h, demonstrating a functional role for *Gm14014* in epicardial cell migration (Figure 4B and Appendix A). Furthermore, phospho-histone 3 (pHH3) immunohistochemical assays demonstrate no significant differences in cell proliferation, supporting the notion that such migratory differences are independent of cell proliferation (Figure 4C). In addition, time lapse analyses revealed that *Gm14014* inhibition impaired cell migration compared to controls, with a notable increase in non-linear migration patterns (i.e., random) (Figure 4D and Appendix A).

To uncover if the functional role of *Gm14014* in migration also occurs in other cardiovascular cell types, scratch assays were also performed in HL1 cardiomyocytes and MEVEC endocardial cells. Importantly, no differences in cell migration were observed during any of the time point analyses (Figure 4B), supporting the notion that *Gm14014’s* role in cell migration is cell-type-specific, i.e., epicardial cells.

Multiple cytoskeletal proteins were identified in our PD assay to putatively interact with *Gm14014*, including those controlling actin filament organization (Rac1), positive and negative actin polymerization (cofilin, profilin, gelsolin, filamin) and acting interacting proteins (actins, actinins, talin, tubulin, myosins, tropomyosin and vinculin). We therefore tested whether differences in epicardial cell migration after *Gm14014* inhibition are induced by changes in the cell cytoskeleton. Immunocytochemical analyses of Actn1, Actn4 and Rac1 displayed no significant differences between *Gm14014* ASO-treated epicardial cells and controls at 6 h (*Gm14014* down-regulated) or 24 h (*Gm14014* up-regulated) (Figure 4E). On the other hand, Myh9 and Myl9 exhibited significant differences at 6 h after ASO administration, since *Gm14014* inhibition significantly reduced the Myl9 and Myh9 protein levels. However, when *Gm14014* was up-regulated at 24 h after ASO treatment, we only observed increased Myh9 protein levels, while for Myl9 there were no significant differences (Figure 4E). Furthermore, no differences in the expression of Myh9 and Myl9 were observed in HL1 atrial cardiomyocytes at both 6 and 24 h, in line with their lack of migration differences following *Gm14014* ASO1 administration (Appendix A). It is important to highlight that, in this context, Myl9 expression is significantly diminished in both nuclear and cytoplasmic compartments while Myh9 only displayed differences at the cytoplasmic level (Figure 4E). Such observations might be linked to the capacity of Myl9 to translocate into the nucleus and exert transcriptional activity [69]. To further sustain the plausible role of *Gm14014* modulating such cytoskeletal proteins, PD assays and WB were performed, confirming that Myl9, but not Rac1, interacted with *Gm14014* (Figure 4F). To provide further evidence of the possible role of *Gm14014* in regulating epicardial cell migration through cytoskeletal protein binding, *Myl9* loss-of-function assays were performed. Two different strategies were employed: *Myl9* siRNA to inhibit its expression in the cytoplasm, and *Myl9* ASO to inhibit its expression, both in the nucleus and cytoplasm (Appendix A). The results were similar to those obtained with *Gm14014* inhibition, showing a reduction in epicardial cell migration capacity from 6 h to 24 h when the *Myl9* expression levels were diminished in both cellular compartments (Figure 4B). Thus, these data demonstrate the involvement of *Gm14014* in the modulation cytoskeletal proteins that play a regulatory role in the epicardial cell migration process.

### 2.9. Dissecting the Functional Role of Wt1_76127

*Gm14014* shares almost 40% nucleotide similarity with *Wt1_76127* (Appendix A). However, it remains unclear if such a sequence conservation is sufficient to maintain similar molecular functional capabilities. To gain insights into the plausible conserved functional role of *Wt1_76127*, we also designed an ASO against this lncRNA. We isolated primary cultures of cardiac embryonic fibroblasts and cardiomyocytes as a proxy to test the plausible functional role of *Wt1_76127*. The expression of *Wt1_76127* was primarily detected in cardiomyocytes (Figure 5A), particularly in the nucleus (Figure 5B) and therefore we selected this cell type for further analyses. Inhibition with 20 nM *Wt1_76127* ASO yielded almost 70–80% inhibition in both cardiomyocytes and epicardial cells (Figure 5C). Cell viability was assayed in CMs, demonstrating that no differences were observed after *Wt1_76127* inhibition, while a small but significant increase in apoptosis was detected (Figure 5D).

To analyze the possible functional conservation between *Gm14014* and *Wt1_76127*, the proliferation and migration of CMs were analyzed after ASO treatment, displaying no significant differences (Figure 5E,G). Curiously, cytoskeletal proteins such as Actn1 and Myh9, but not Actn4, were significantly down-regulated (Figure 5J). Additional experiments using epicardial explants assays [70,71,72] demonstrated neither significant differences in cell migration (Figure 5H) nor in Actn1, Actn4 or Myh9 immunohistochemical detection in migrating epicardial cells (Figure 5I). Therefore, these data demonstrated that *Wt1_76127* inhibition does not influence epicardial and/or myocardial cell migration in chicken embryos and suggested that the function of this lncRNA is not conserved across species.

To further explore the plausible role of *Wt1_76127* during embryonic development, in vivo pericardial injections were performed in HH17 chicken embryos (Appendix A). Viability was not compromised by intrapericardial injections of *Wt1_76127* ASO (Appendix A). Similarly, neither cardiac rhythm, i.e., beats per minute and beating regularity, and cardiac development were compromised (Appendix A). To further confirm the in vitro data obtained by *Wt1_76127* silencing, we isolated and ex vivo cultured ventricular explants after ASO administration and the migration capacity of epicardial cells was analyzed. No significant differences in epicardial cell migration between control and ASO-treated explants were observed (Appendix A), supporting the notion that the function of *Wt1_76127* and *Gm14014* lncRNAs is not conserved between species.

## 3. Discussion

LncRNAs represent a large family of non-coding RNAs with a wide cellular and tissue distribution. We have previously characterized three newly identified lncRNAs in the chicken genome that display an enhanced expression in the developing PE as compared to the embryonic ventricle [55]. We have now provided further insight into the distinct characteristics of these lncRNAs, including a more detailed analysis of their tissue distribution, transcriptional regulation, functional role and evolutionary conservation in mice. Our data demonstrated that these lncRNAs display a broad tissue distribution, in line with other reports demonstrating the global expression of lncRNAs in different tissues [56,73,74]. In addition, we also demonstrated that these lncRNAs are preferentially located in the nucleus, supporting in the first instance a more plausible role in transcriptional and/or epigenetic regulation [56]. Intriguingly, their subcellular localization is modulated as embryonic development advances in a tissue-specific manner, i.e., more prominent in the embryonic ventricles as compared to the PE/EE. Thus, these observations support the notion of a dynamic lncRNA subcellular localization that might reflect changes in their maturation and/or biological function [75], a process that seems to be distinctly regulated by transcription factors such as *Mef2c*, as demonstrated herein.

Cardiac-enriched transcription factors such as *Mef2c* [76,77,78], *Nkx2.5* [79,80,81], *Srf* [82,83] and *Pitx2* [84,85,86,87] play pivotal roles in cardiogenesis as well as in proepicardial/embryonic epicardial development [25,26,88,89] by transcriptionally regulating multiple downstream targets, particularly in heart development. While ample evidence is reported on the transcriptional regulation of protein-coding RNA [90,91,92,93], scarce evidence is available regarding non-coding RNAs, particularly lncRNAs [62,94,95,96]. Herein, we provide evidence that *Mef2c* exerts a robust transcriptional repression in embryonic ventricle but not in the epicardium (except for *Wt1_76127*) while such repressive signals are mostly mediated by *Nkx2.5* in the epicardium. On the other hand, *Pitx2c* and *Srf* are essential for proper lncRNAs expression in the epicardium, whereas they are mostly dispensable in the embryonic ventricle. Curiously, both the *Mef2c* gain- and loss-of-function assays resulted in a significant up-regulation of *Gm14014* in epicardial cells (MEC1). The precise molecular mechanisms leading to a similar output expression remains to be established; however, there are several plausible explanations that can lead to equal lncRNAs deregulation. For example, there might be a transcriptional negative feedback loop, a limiting cofactor or an impaired activator/repressor equilibrium regulated by *Mef2c* that can result in both cases (over-expression and inhibition) occurring in the same gene target expression output. Overall, these data reinforce the notion that lncRNA transcriptional expression is modulated by distinct cardiac-enriched transcription factors, as previously reported for other lncRNAs [94,95] in a tissue-specific manner.

Long non-coding RNAs are poorly conserved non-coding RNA molecules [48,62], in contrast to microRNAs [47]. We could only identify a mouse homologue for *Wt1_76127* in the mouse genome given that it is located in the syntenic genomic locus, i.e., the vicinity of *Wt1* transcription factor and shares a 39.1% nucleotide homology. Analyses of its embryonic and adult tissue distribution as well as its subcellular distribution within distinct cardiovascular cellular types display a rather similar profile as its chicken homologue, further supporting its evolutionary conservation.

On the other hand, the transcriptional regulation of *Gm14014* displays a mixture of similar control mechanisms to the chicken *Wt1_76127* homologue, as *Nkx2.5* and *Srf* are completely dispensable for their expression in myocardial cells while *Mef2c* and *Pitx2c* exert repressive and activating conditions on epicardial cells. On the other hand, divergent roles for *Mef2c* and *Pitx2c* are observed in myocardial cells, displaying opposite transcriptional regulation in the chicken vs. the mouse model. Thus, these data support the notion that cardiac-enriched transcription factors can distinctly and selectively regulate homologue lncRNA expression in a cell-specific manner, as previously reported for other lncRNAs [94,95,97]. Furthermore, the transcriptional regulation of lncRNAs seems to have evolved rapidly among different species, as previously reported [48,62], supporting the notion that they might even exert divergent functional roles in chicken vs. mice. Thus, these data support the notion that a limited extrapolation to the transcriptional regulation of lncRNAs between species could be performed.

LncRNAs can modulate multiple biological processes, including epigenetic, transcriptional and post-transcriptional processes [48,62,75]. Dissecting their subcellular localization and searching for protein partners can provide further evidence of their functional roles [48,62,75]. Herein, we demonstrated through RT-qPCR as well as SCRINSHOT that *Gm14014* is widely expressed in different tissues and cell types. Within the cardiovascular context, its expression is prominently observed in MEC1 epicardial and MEVEC endocardial cells as compared to HL1 myocardial cells and its subcellular localization in all these cell types is prominently nuclear. Several lncRNAs have been reported to play pivotal roles during cardiogenesis, such as HBL1 [98,99], Linc1405 [100], CARMA [101], Moshe [102] and Braveheart [49], during cardiac diseases such as Sweetheart [103], lncExACT1 [104] and Malat1 [105] as well as promoting cardiac regeneration such as CAREL [106] and Snhg1 [107] [for a recent review, see [108]]. Importantly, to the best of our knowledge, the role of lncRNAs has not been reported during epicardial development in any experimental model, highlighting the novelty of our work.

In addition, functional in vitro analyses demonstrated that *Gm14014* is essential for epicardial cell migration, but not for endocardial and myocardial migration, supporting its cell-type-specific role. A large body of evidence has been reported on the role of lncRNA in oncogenic cell migration and metastasis (for recent reviews, see [109,110,111]], but, to the best of our knowledge, this is the first evidence of the functional role of an lncRNA in homeostatic cell migration, with implications for embryonic epicardial development.

Mechanistically, the identification of *Gm14014*-associated proteins by MS revealed a similar number of nuclear, cytoplasmic and nuclear/cytoplasmic proteins. Silencing assays demonstrate that several cytoskeletal proteins such as Myh9 and Myl9 are severely impaired, supporting a plausible molecular link to impaired epicardial cell migration [69,112,113,114]. However, it remains to be reconciled why *Gm14014* is primarily localized in the nucleus but cytoskeletal reorganization is observed. A plausible explanation is that immature *Gm14014* transcripts are retained in the nucleus and only a subset is selectively translocated towards the cytoplasm to exert its functional role [75], a process that might be developmentally and transcriptionally dynamic. This is, indeed, in line with the fact that chicken *Gm14014* homologue lncRNA subcellular expression, i.e., *Wt1_76127*, during epicardial and myocardial maturation, becomes progressively more abundantly expressed in the cytoplasm as development proceeds. A similar observation has also been reported for other lncRNAs during development [26].

Alternatively, *Gm14014* can interact with nuclear proteins that transcriptionally control the expression of key master genes regulating actin cytoskeletal proteins, since multiple nuclear proteins were also detected in our MS analyses. We tested whether Rac1 might be this linking factor, but failed to demonstrate a physical *Gm14014* interaction or immunocytochemical differences. Thus, it might be plausible that *Gm14014* might be interacting with proteins that are both nuclear and cytoplasmic. We provided evidence that *Gm14014* physically interacts with Myl9. Furthermore, we demonstrated that the inhibition of *Gm14014* selectively translocates Myl9 to the cytoplasm and additionally diminishes *Myh9* expression, a down-regulation that is selectively observed only in MEC1 epicardial cells where migration is also halted by *Gm14014* silencing but not in other cell types. Therefore, these observations support a working model in which *Gm14014* physically interacts with Myl9 in the nucleus, facilitating *Myh9* transcription and thus modulating cell migration, as previously reported [69] (Figure 6). In the absence of *Gm14014*, Myl9 translocates to the cytoplasm and thus no longer promotes *Myh9* transcription. As a consequence, *Myh9* is down-regulated in the cytoplasm, leading to cytoskeletal remodeling and thus halting cell migration (Figure 6). Surprisingly, such molecular mechanisms are not observed in the chicken *Gm14014* homologue, i.e., *Wt1_76127*, demonstrating divergent functional roles.

In summary, herein, we provide evidence that chicken *Wt1_76127* lncRNA and its murine homologue, i.e., *Gm14014*, display similar tissue distribution profiles in the embryonic and adult stages. However, transcriptional regulation by key cardiac-enriched transcription factors displays conserved and divergent profiles. Additionally, we demonstrated that murine *Gm14014*, but not chicken *Wt1_76127,* is essential for epicardial, but not myocardial, cell migration, a process that is modulated by physical *Gm14014*-Myl9 interactions and subsequent nuclear-to-cytoplasmic translocation and cytoskeletal rearrangement. Thus, these data support the notion that homologues lncRNAs can exert distinct and species-specific functional capabilities and open future research avenues to explore the role of these lncRNAs in cardiac injury and regeneration given the prominent role of the epicardium in these biological contexts.

## 4. Methods

### 4.1. Chicken Embryonic Tissues and Epicardial Explants

Fertilized eggs from white Leghorn chickens (Granja Santa Isabel, Córdoba, Spain) were incubated at 37.5 °C and 50% humidity for 2–7 days. Embryos were harvested at different developmental stages (HH17, HH24 and HH32) and classified according to Hamburger and Hamilton [115]. Embryos were removed from the egg by cutting the blastocyst margin with iridectomy scissors and placed into phosphate-buffered saline (PBS). For RT-qPCR analyses, HH17 PEs were dissected, pooled (n = 10) and stored at −80 °C until use. Additionally, in vitro explants cultures and HH24 and HH32 cardiac explants were cultured for 24 h and subsequently the ventricular and epicardial outgrowths were separated, isolated, pooled (n = 10) and stored at −80 °C until use.

### 4.2. Chicken Primary Cultures

HH36 embryonic hearts were isolated and disintegrated with iridectomy scissors and placed in PBS. The tissues were incubated with trypsin for 30 min at 37 °C and the supernatant was collected. This trypsinization step was repeated until all the tissue was fully dissociated. The supernatant was centrifuged and the pellets were cultured in plastic Petri dishes with fibroblast culture medium (Dulbecco’s Modified Eagle’s Medium-high glucose) (Sigma, Munich, Germany) supplemented with fetal bovine serum (FBS) 10%. Two pre-plating steps were carried out to separate cardiac fibroblasts (CFs) and cardiomyocytes (CMs). Subsequently, CMs from the supernatant were cultured in a plastic Petri dish with s fresh CM culture medium supplemented with 0.001 M 5-bromo-2-deoxyuridine (Sigma) to inhibit fibroblast proliferation as previously reported [116].

### 4.3. Mouse Embryonic Tissues

CD1 mice were bred and embryos were collected at different embryonic developmental stages, including embryonic days (E) E10.5, E13.5, E16.5 and E19.5. Postnatal hearts (from days P2, P7 and P21) were also collected. Pregnant females and neonatal mice were euthanized by cervical dislocation and by decapitation, respectively. Subsequently, the embryonic and postnatal tissues were dissected, pooled and stored at −80 °C until use. Approval was obtained from the Andalusian Ethic Committee prior to the initiation of the study.

### 4.4. Nucleus/Cytoplasm Subcellular Isolation

Cytoplasmic and nuclear RNA fractions from HH17 PE, HH24 and HH32 epicardial/myocardial outgrowths as well as from primary chicken cardiomyocytes, HL1 cardiomyocytes (SCC065, Sigma-Aldrich), MEVEC endocardial cells [117], MEC1 (SCC187, Sigma-Aldrich) and EPIC epicardial cells [118] were isolated with the Cytoplasmic and Nuclear RNA Purification Kit (Cat. 21000, Norgen, Belmont, CA, USA) following the manufacturer’s instructions. The kit allowed for the isolation of nuclear and cytoplasmic RNA from other cellular components such as genomic DNA or proteins. The method used for RNA isolation was rotary column chromatography using Norgen resin as a separation matrix. After RNA isolation, RT-qPCR analyses for nuclear-enriched *Xist* (isoform 2) mRNA and cytoplasmic *Gapdh* mRNA were performed to validate their enrichment on each subcellular fraction. An RT-qPCR analysis of distinct lncRNAs was subsequently performed as detailed below.

### 4.5. siRNA Cell Transfections

Chicken ventricular and epicardial explants, HL1 cardiomyocytes, MEVEC endocardial cells, MEC1 and EPIC epicardial cells (6 × 10^5^ cells per well) were transfected with Pitx2c-siRNA, Srf-siRNA, Nkx2.5-siRNA, Mef2c-siRNA and Myl9-siRNA (Sigma, Aldrich, Munich, Germany), respectively, as previously described [119,120]. siRNA sequences are provided in Appendix A. Validation of the siRNAs’ inhibition was carried out by RT-qPCR assays.

### 4.6. ASO Design and Transfection

Antisense oligonucleotides (ASOs) were designed as previously reported [121]. The structure of ASOs used in this report consisted of a main backbone (10 nucleotides) with phosphorothioate groups, and 5 nucleotides on both sides with different methyl groups. ASO transfections were carried out with Lipofectamine 2000 (Invitrogen, Carlsbad, CA, USA) following the manufacturer’s guidelines. Concentrations of 20 nM of *Wt1_76127* and *Gm14014* ASO were applied to the cells at different times, respectively, while for *Myl9* ASO, the concentration used was 80 nM. The ASO sequences are provided in Appendix A. Validation of the ASOs treatment was carried out by RT-qPCR assays.

### 4.7. Cell Viability Assays

Cell viability was analyzed with an Apoptosis/Necrosis Assay Kit (Abcam, Cambridge, UK), following the manufacturer’s instructions. Cell cultures were analyzed using a Leica TCS SP5 II confocal scanning laser microscope. Viable cells were detected with Cytocalcein violet 450 reagent (blue), apoptotic cells with Apopxin Deep Red (red) and apoptotic and necrotic cells were detected with Nuclear Green DCS1 reagent (green).

### 4.8. Cell Migration Assays

Cell migration was analyzed using the scratch assay, as described by Ascione et al. [122]. Primary chicken cardiomyocytes, HL1 cardiomyocytes, MEVEC endocardial cells and MEC1 epicardial cells were plated on 24-well culture dishes at a density of 6 × 10^5^ cells per well and incubated until 90–100% confluence. Cell monolayers were manually scratched with a p200 pipette tip. PBS was used to wash the cells and was subsequently replaced with the serum-free medium. The experimental group was treated with lipidic vesicles containing ASOs as cargo, while the control group was treated with empty lipidic vesicles, respectively. All plates were photographed after 24 h. In addition, a time lapse analysis was carried out for the two conditions. Cell monolayers were scratched, transfected and cultured for 24 h with images taken every 15 min. A Leica TCS SP5 II confocal microscope was used, maintaining optimal temperature and humidity conditions for cell growth.

### 4.9. Growth Factors and Thymosin β4 Administration

HL1 cardiomyocytes, MEVEC endocardial cells and MEC1 epicardial cells were treated for 24 h with Bmp2, Bmp4, Fgf2, Fgf8 and thymosin *β*4 (Prospec, East Brunswick, NJ, USA), respectively, as reported by Dueñas et al. [55]. Cells were collected and processed accordingly for RT-qPCR. In all cases, 3–5 independent biological replicates were analyzed.

### 4.10. Immunohistochemical Analyses

Control and experimentally treated cells were collected after the corresponding treatment, rinsed in PBS for 10 min and fixed with 4% PFA for 10 min at room temperature. After fixation, the samples were rinsed three times (10 min each) in PBS at room temperature and then permeabilized with 1% Triton X-100 and NH_4_Cl 50nM in PBS for 10 min at room temperature. To block nonspecific binding sites, PBS containing gelatin solution 0.2% (Sigma) was applied twice (10 min each). After blocking, the samples were rinsed three times (10 min each) in PBS at room temperature and were immunofluorescently labeled to detect different proteins. Primary antibodies against Actn1 (ab68194, Abcam), Actn4 (ab108198, Abcam), Myh9 (ab75590, Abcam), pHH3 (CA-92590, Milipore, Burlington, MA, USA), Myl9 (sc-28329, Santa Cruz, Dallas, TX, USA), Rac1 (sc-514583, Santa Cruz) and MF20 (ATCC) were used, which were diluted (1:200) in blocking solution and applied to each culture overnight at 4 °C. Subsequently, the samples were rinsed three times (10 min each) in PBS to remove the excess primary antibodies and incubated at room temperature for 30 min with Alexa-Fluor 546, 488 and 633 (1:100; Invitrogen) as a secondary antibody, respectively. Negative controls were produced without incubation with the corresponding specific primary antibody, and no signals were obtained for any of them after incubation with the secondary antibody. Finally, the cells were incubated with DAPI (1:2000; Sigma) for 10 min at room temperature and rinsed three times in PBS for 5 min each. Cell cultures were stored in PBS in darkness at 4 °C until they were analyzed using a Leica TCS SP5 II (Wetzlar, Germany) confocal scanning laser microscope.

### 4.11. SCRINSHOT In Situ Hybridization

The SCRINSHOT (*Single-Cell Resolution*
IN Situ
*Hybridization On Tissues*) assay was conducted as described by Sountoulidis et al. [123]. Heart tissues from mouse embryos (E10.5, E13.5, E16.5, E19.5) and 2-, 7- and 21-day-old (P2, P7 P21) animals were collected, washed in PBS 1X (pH = 7.4), embedded in OCT (FSC22 Clear, Leica, Wetzlar, Germany) and stored at −80 °C until used. Then, 5 µm thick cryostat sections were obtained (Leica CM3050S). Padlock probes (40–45 nucleotides) were designed using the PrimerQuest online tool (IDT: Integrated DNA Technologies, Park Coralville, IA, USA). These DNA oligos were used to design the fluorophore detection oligos, replacing 2–3 “T” nucleotides with “U” nucleotides to subsequently remove these oligos after the detection cycle using the enzyme Uracil-DNA Glycosylase (Thermo, EN0362, Waltham, MA, USA). The sequences of the padlock probes and the fluorophore detection oligos are provided in Appendix A. Images were obtained using a Zeiss Axio Observer Z.2 fluorescent microscope (Oberkochen, Germany). The image analysis was based on the nuclear segmentation and alignment of the different detection cycles for the genes of interest. Colocalization was obtained as the number of positive cells for the gene of interest relative to the total number of cells in each sample. Further image processing was carried out using FIJI 2.9.0, Cell Profiler 4.2.5, RStudio 2024.04.2 and TissUUmaps 3.0.10.1 software.

### 4.12. RNA Isolation and cDNA Synthesis

Total RNA was isolated using the Direct-Zol RNA Miniprep Kit (Zymo Research, Irvine, CA, USA), according to the manufacturer’s instructions. In all cases, at least three distinct pooled samples were used to perform the corresponding RT-qPCR experiments. For mRNA expression measurements, the reverse transcription Maxima First Strand cDNA Synthesis Kit for RT-qPCR (Thermo Scientific) was used, according to the manufacturer’s guidelines. Negative controls to assess genomic contamination were generated for each sample, without reverse transcriptase, which resulted in all cases in no detectable amplification product.

### 4.13. qPCR Analyses (mRNA and lncRNA)

Real-time PCR experiments were performed with 2 μL of diluted cDNA, GoTaq^®^ qPCR Master Mix (Promega, Madison, WI, USA) and the corresponding primer sets described in Appendix A. All qPCRs were performed using a CFX384TM thermocycler (Bio-Rad, Hercules, CA, USA) following the manufacturer’s recommendations. The relative level of expression of each gene was calculated as described by Livak and Schmittgen [124] using *Gapdh* as an internal control for mRNA expression analyses. All primers were designed to span exon–exon boundaries using the online Primer3 software Primer3Plus (https://www.bioinformatics.nl/cgi-bin/primer3plus/primer3plus.cgi). Each PCR reaction was carried out in triplicate and repeated with at least three distinct biological samples to obtain representative means. No amplifications were observed in PCR negative control reactions containing only water.

### 4.14. LncRNA Pull Down Assays

Pull down of biotinylated RNA was carried out as described by Panda et al. [125]. The biotinylated RNA of exon 1 and exon 2 of *Gm14014* and *Gapdh* were synthesized from PCR fragments using specific forward primers that contained the T7 RNA polymerase promoter sequence [(T7) AGTAATACGACTCACTATAGGG]. Seven fragments were obtained for the sequence of *Gm14014* and one fragment for *Gapdh*. The fragments were biotinylated with Biotin-14-CTP (Thermo Fisher Scientific, Invitrogen) and transcribed with the MaxiScript T7 kit (Thermo Fisher Scientific, Invitrogen). Whole-cell lysates (500 μg) from MEC1 cells were incubated with 1 μg of biotinylated RNA (biotinylated *Gm14014* and *Gapdh* samples) for 2h at room temperature. An input sample incubated only with cell lysate was included as a negative control. Complexes were isolated with Streptavidin-coupled Dynabeads (Invitrogen) and analyzed by mass spectrometry (MS). DAVID database and Gene Ontology analyses were subsequently performed on the resulting proteomic data.

### 4.15. Western Blot

Western blot (WB) was performed with 10% of the pull-down lysate to validate the interaction between *Gm14014*, Myl9 and Rac1. The primary antibodies Myl9 (sc-28329, Santa Cruz) and Rac1 (sc-514583, Santa Cruz) were used at a concentration of 1:100 and incubated for 5h at room temperature, and the secondary antibody–HRP conjugate (170-6516, Biorad) at 1:5000 for 30 min at room temperature. Blocking was carried out with albumin and washes were performed with PBST, according to the antibody manufacturer’s recommendations.

### 4.16. Statistical Analyses

For the statistical analyses of datasets, unpaired Student’s t-tests were used with a 95% confidence interval. The significance levels of the *p*-values are stated in each corresponding figure legend: * *p*-value < 0.05; ** *p*-value < 0.01; *** *p*-value < 0.001; **** *p*-value < 0.0001. GraphPad Prism (8.0.2) software was used for the statistical analysis and graphical representation.

## 5. Conclusions

Herein, we provided evidence that chicken *Wt1_*76127 and its mouse homologue *Gm14014* are widely distributed in different embryonic and adult tissues, displaying a prominent nuclear localization. The transcriptional regulation of these lncRNAs is exerted by distinct cardiac-enriched transcription factors, revealing divergent functionalities in chicken and mice. Mechanistically, *Gm14014* is required for epicardial cell migration, a process mediated by partnering with Myl9, but not *Wt1_76127*, highlighting evolutionary differences. Thus, our data demonstrate the pivotal role of *Gm14014* lncRNA in cell migration, a biological process that is crucial during cardiac regeneration.

## Figures and Tables

**Figure 1 ijms-25-12904-f001:**
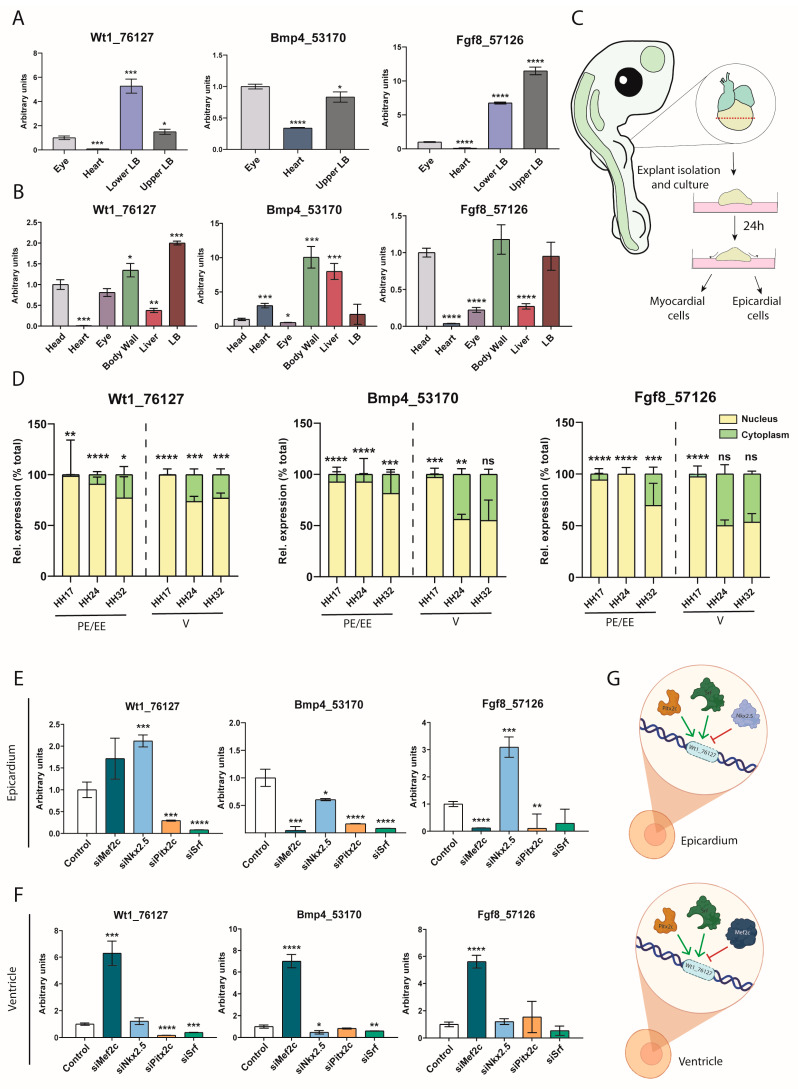
***Wt1_76127*, *Bmp4_53170* and *Fgf8_57126* are predominant nuclear lncRNAs, exhibiting ubiquitous expression in multiple tissues and being regulated by different cardiac enriched transcription factors.** Panel (**A**). RT-qPCR analyses of *Wt1_76127*, *Bmp4_53170* and *Fgf8_57126* in HH24 embryonic tissues, demonstrating high expression levels in the limb buds compared to the heart (n = 3; each biological sample (n) corresponds to 3–5 pooled tissues). Panel (**B**). RT-qPCR analyses of *Wt1_76127*, *Bmp4_53170* and *Fgf8_57126* in HH32 embryonic tissues, demonstrating high expression levels in the limb buds, body wall and liver compared to the heart and eye (n = 3; each biological sample (n) corresponds to 3–5 pooled tissues). Panel (**C**). Schematic representation of the chicken ventricular explants. Panel (**D**). RT-qPCR analyses of the subcellular nuclear and cytoplasmic distribution of *Wt1_76127*, *Bmp4_53170* and *Fgf8_57126* at HH17, HH24 and HH32 in proepicardial (PE; HH17), embryonic epicardial (EE, HH24 and HH32) and ventricular (V, HH17, HH24 and HH32) tissues. It can be observed that all three lncRNAs are prominently nuclear but their cytoplasmic expression increases as development proceeds, particularly in the ventricular tissues (n = 3; each biological sample (n) corresponds to 3–5 pooled tissues). Panel (**E**). RT-qPCR analyses of *Wt1_76127*, *Bmp4_53170* and *Fgf8_57126* expression in epicardial cells after selective inhibition of *Mef2c*, *Nkx2.5*, *Pitx2c* and *Srf* transcription factors by siRNA administration (n = 3; each biological sample (n) corresponds to a single transfection assay). Panel (**F**). RT-qPCR analyses of *Wt1_76127*, *Bmp4_53170* and *Fgf8_57126* expression in ventricular cells after selective inhibition of *Mef2c*, *Nkx2.5*, *Pitx2c* and *Srf* transcription factors by siRNA administration (n = 3; each biological sample (n) corresponds to a single transfection assay). Panel (**G**). Schematic representation of the transcriptional regulation of *Wt1_76127* in epicardial and ventricular samples. It can be observed that *Nkx2.5* exerts transcriptional regulation in the epicardium, *Mef2c* modulates *Wt1_76127* in the ventricle while *Pitx2c* and *Srf* exert a transcriptional role in both cell types. Statistical analysis: t-student (95% confidence interval); * *p*-value < 0.05; ** *p*-value < 0.01; *** *p*-value < 0.001; **** *p*-value < 0.0001. ns, not significant. Figures were made with Biorender (https://www.biorender.com) and Adobe Illustrator CC, 23.01.

**Figure 2 ijms-25-12904-f002:**
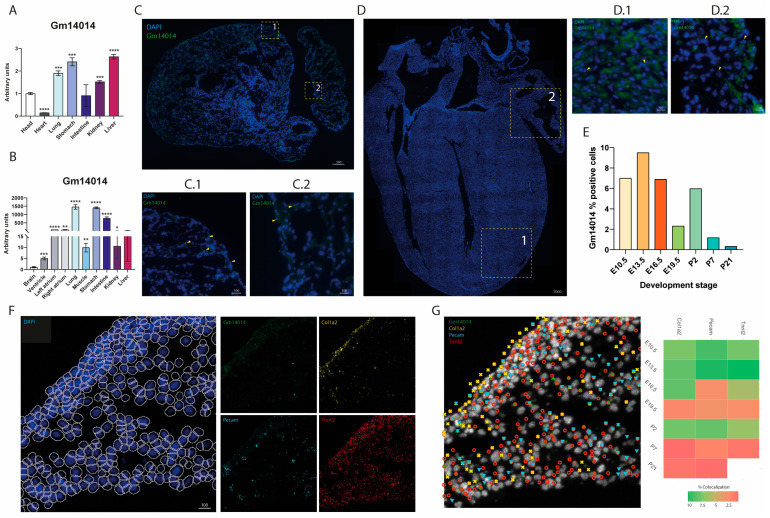
***Gm14014* is expressed in multiple embryonic and adult tissues, with higher expression during the embryonic stages.** Panel (**A**). RT-qPCR analyses of *Gm14014* expression in embryonic (E13.5) mouse tissues, demonstrating a wide expression in different tissues, being higher in the lung, stomach and liver compared to other tissues, e.g., heart. n = 3; each biological sample (n) corresponds to 5–7 pooled tissues). Panel (**B**). RT-qPCR analyses of *Gm14014* expression in adult (3 months old) mouse tissues, demonstrating a wide expression in different tissues, being higher in the lung, stomach and liver compared to other tissues, e.g., heart. n = 3; each biological sample (n) corresponds to 3 pooled tissues). Panel (**C**). SCRINSHOT in situ hybridization analyses of *Gm14014* expression in E13.5 mouse hearts; panels C.1 and C.2 are close ups corresponding to atrial and ventricular areas, respectively (n = 3). Expression is demarcated by yellow arrows. Panel (**D**). SCRINSHOT in situ hybridization analyses of *Gm14014* expression in 21 days postnatal (P21) mouse hearts; panels D.1 and D.2 are close ups corresponding to atrial and ventricular areas, respectively (n = 3). Expression as demarcated by yellow arrows. Panel (**E**). SCRINSHOT in situ hybridization analyses of *Gm14014*-positive cells in embryonic (E10.5, E13.5, E16.5, E19.5) and postnatal (P2, P7, P21) hearts. Panel (**F**). SCRINSHOT in situ hybridization analyses with nuclei segmentation allowing us to discern the colocalization of *Gm14014*, *Col1a2*, *Pecam* and *Tnnt2* expression, respectively. These images correspond to an E13.5 heart. Panel (**G**). SCRINSHOT in situ hybridization analyses of *Gm14014, Col1a2*, *Pecam* and *Tnnt2* demarcating their specific cellular localization. On the right is a heatmap displaying the rate of *Gm14014* colocalization with *Col1a2, Pecam* and *Tnnt2, respectively*. Note that increased colocalization during the early embryonic stages as well as the postnatal P2 stage (green) was observed, while the lowest rate was observed during postnatal stages P7 and P21 (red). Statistical analysis: Student’s t (95% confidence interval); * *p*-value < 0.05; ** *p*-value < 0.01; *** *p*-value < 0.001; **** *p*-value < 0.0001. Schemes were made with TissUUmaps 3.0.10.1.

**Figure 3 ijms-25-12904-f003:**
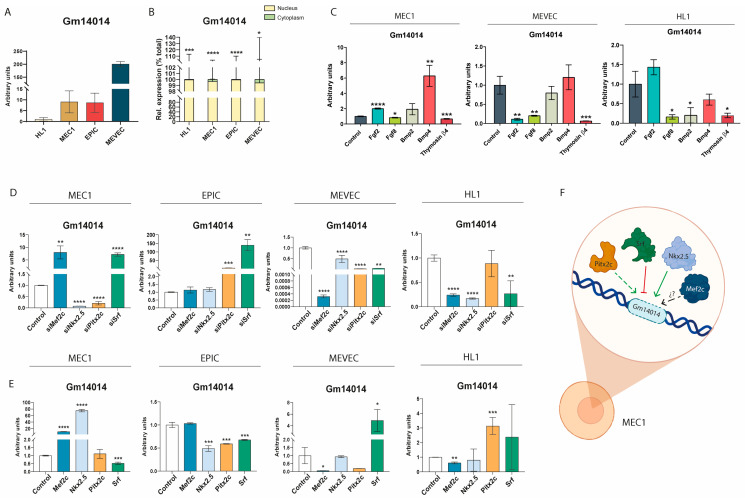
**Nuclear *Gm14014* is differentially expressed in distinct cardiac cell lines and differently regulated by growth factors and cardiac enriched transcription factors.** Panel (**A**). RT-qPCR analyses of *Gm14014* expression in HL1 cardiomyocytes, MEC1 and EPIC epicardial cells, and MEVEC endocardial cells. Higher expression levels were observed in MEVEC endocardial cells as compared to MEC1 and EPIC epicardial cells, with lower expression levels in HL1 cardiomyocytes (n = 3; each biological sample (n) corresponds to a single transfection assay). Panel (**B**). RT-qPCR analyses of the subcellular nuclear and cytoplasmic distribution of *Gm14014* in HL1 cardiomyocytes, MEC1 and EPIC epicardial cells, and MEVEC endocardial cells. Note that in all cases, *Gm14014* is prominently nuclear. Panel (**C**). RT-qPCR analyses of *Gm14014* expression in MEC1 epicardial cells, MEVEC endocardial cells and HL1 cardiomyocytes after the selective administration of *Fgf2*, *Fgf8*, *Bmp2*, *Bmp4* and thymosin β4, respectively (n = 3l each biological sample (n) corresponds to a single transfection assay). Panel (**D**). RT-qPCR analyses of *Gm14014* expression in MEC1 and EPIC epicardial cells, MEVEC endocardial cells and HL1 cardiomyocytes after the selective inhibition of *Mef2c*, *Nkx2.5*, *Pitx2c* and *Srf* transcription factors by siRNA administration, respectively (n = 3; each biological sample (n) corresponds to a single transfection assay). Panel (**E**). RT-qPCR analyses of *Gm14014* expression in MEC1 and EPIC epicardial cells, MEVEC endocardial cells and HL1 cardiomyocytes after the selective over-expression of *Mef2c*, *Nkx2.5*, *Pitx2c* and *Srf* transcription factors, respectively (n = 3; each biological sample (n) corresponds to a single transfection assay). Panel (**F**). Schematic representation of the transcriptional regulation of *Gm14014* in MEC1 epicardial cells. It can be observed that *Nkx2.5* and *Srf* display opposite regulatory patterns while *Pitx2c* inhibition up-regulates *Gm14014*. Statistical analysis: Student’s t (95% confidence interval); * *p*-value < 0.05; ** *p*-value < 0.01; *** *p*-value < 0.001; **** *p*-value < 0.0001. Schemes were made with Biorender.

**Figure 4 ijms-25-12904-f004:**
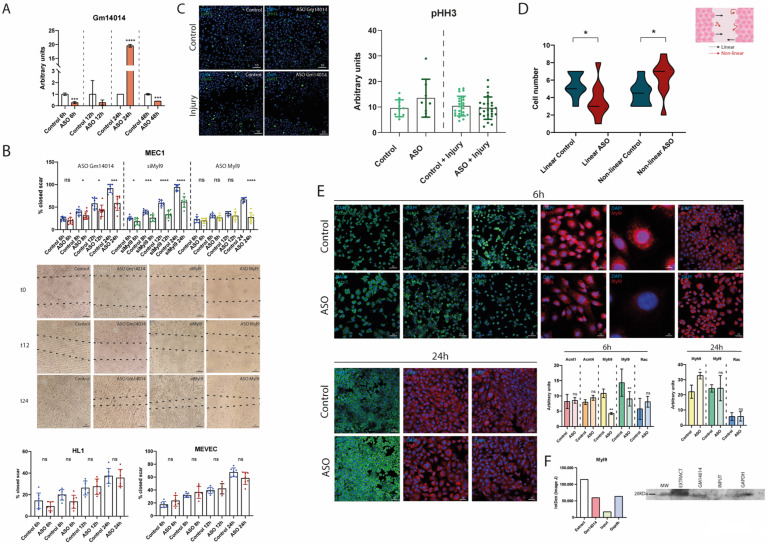
***Gm14014* regulates epicardial cell migration by modulating cytoskeletal proteins.** Panel (**A**). RT-qPCR analyses of *Gm14014* expression in MEC1 epicardial cells after *Gm14014* ASO administration 6 h, 12 h, 24 h and 48 h after transfection. It can be observed that selective down-regulation was achieved at 6 h, 12 h and 48 h while at 24 h there was a significant up-regulation (n = 3). Panel (**B**). Schematic representation of wound healing scratch assay in MEC1 epicardial cells in controls, *Gm14014* ASO-, siMyl9- and *Myl9* ASO-treated cells, HL1 cardiomyocytes and MEVEC endocardial cells at 6 h, 8 h, 12 h and 24 h in controls and *Gm14014* ASO-treated cells. Representative images of MEC1 epicardial cells at t = 0, t = 12 h and t = 24 h. Note that the migration of *Gm14014*-ASO treated cells was significantly decreased in MEC1 epicardial cells but not in HL1 cardiomyocytes or MEVEC endocardial cells. The migration of si*Myl9* and *Myl9* ASO treated cells was significantly decreased in MEC1 epicardial cells. Panel (**C**). Representative images of proliferation assays as revealed by phospho-histone 3 (pHH3) immunocytochemistry in control and scratched MEC1 epicardial cells corresponding to control and *Gm14014* ASO conditions. Quantitative analyses are also shown, demonstrating no significant differences in cell proliferation. Panel (**D**). Graphical representation of lineal vs. non-linear cell migration in time lapse confocal image analyses of control and scratched MEC1 epicardial cells corresponding to control and *Gm14014* ASO conditions. Panel (**E**). Representative images of immunocytochemical analyses of Actn1, Actn4, Myh9, Myl9, Rac1 at 6 h and of Myh9, Myl9, Rac1 at 24 h after administration of *Gm14014* ASO compared to controls. It can be observed that there is significant difference in the expression of Myh9 and Myl9 at 6 h but not at 24 h. Note also that Myl9 displays both nuclear and cytoplasmic distribution in controls while in *Gm14014*-treated cells it is exclusively cytoplasmic. Panel (**F**). Quantitative analysis and representative blot of Myl9 after *Gm14014* PD. Observe that Myl9 protein interacts with Gm14014. Statistical analysis: Student’s t (95% confidence interval); * *p*-value < 0.05; ** *p*-value < 0.01; *** *p*-value < 0.001; **** *p*-value < 0.0001. ns, not significant. Schemes were made with Biorender.

**Figure 5 ijms-25-12904-f005:**
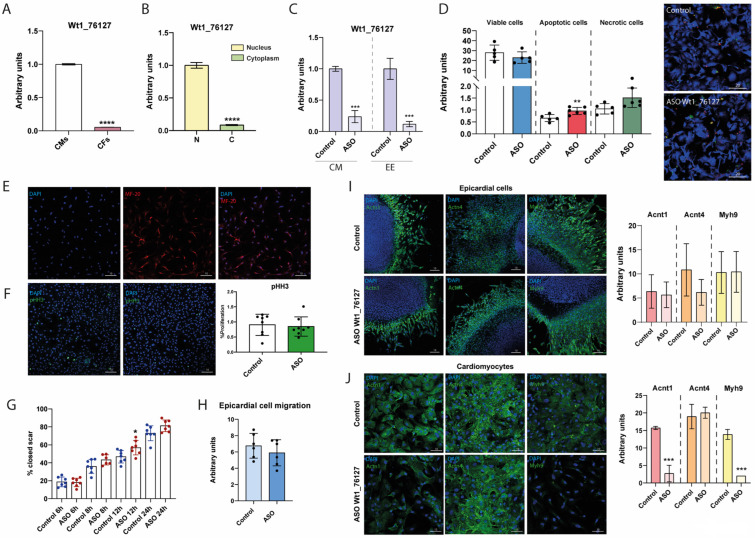
**The functional role of mouse *Gm14014* lncRNA in epicardial cell migration is not conserved by chicken *Wt1_76127* lncRNA.** Panel (**A**). RT-qPCR expression analyses of *Wt1_76127* in primary cultures of chicken embryonic cardiomyocytes (CMs) and cardiac fibroblasts (CFs) (n = 3). Panel (**B**). Subcellular distribution of *Wt1_76127* in the nucleus and cytoplasm primary cultures of chicken embryonic cardiomyocytes (n = 3). Panel (**C**). RT-qPCR expression analyses of *Wt1_76127* in embryonic cardiomyocytes and embryonic epicardium (EE) in control and *Wt1_76127* ASO treated conditions. Observe a significant down-regulation in *Wt1_76127* after ASO administration (n = 3). Panel (**D**). Quantitative analyses of cell viability, apoptosis and necrosis in primary cultures of chicken embryonic cardiomyocytes (CMs) in control and *Wt1_76127* ASO-treated conditions. Representative images are also depicted. Note that there are no significant differences for cell viability and necrosis, whereas there are significant differences in apoptosis. Panel (**E**). Immunocytochemical characterization of a primary culture of embryonic cardiomyocytes with MF20 antibody, demonstrating that most of these cells display sarcomeric myosins. Panel (**F**). Immunocytochemical characterization of proliferation in a scratched primary culture of embryonic cardiomyocytes. Observe that there are no significant differences. Panel (**G**). Schematic representation of a wound healing scratch assay in primary culture of embryonic cardiomyocytes at 6 h, 8 h, 12 h and 24 h in controls and *Wt1_76127* ASO-treated cells, respectively. Panel (**H**). Quantitative analyses of epicardial cell migration in ventricular explant in control and *Wt1_76127* ASO treated conditions. It can be seen that there are no significant differences. Panel (**I**). Representative images of ventricular explants immunostained against Actn1, Actn4 and Myh9 in control and *Wt1_76127* ASO-treated conditions. It can be seen that there are no significant differences. Panel (**J**). Representative images of primary culture of embryonic cardiomyocytes immunostained against Actn1, Actn4 and Myh9 in control and *Wt1_76127* ASO treated conditions. It can be seen that Actn1 and Myh9, but not Actn4, display significant differences. Statistical analysis: Student’s t (95% confidence interval); * *p*-value < 0.05; ** *p*-value < 0.01; *** *p*-value < 0.001; **** *p*-value < 0.0001.

**Figure 6 ijms-25-12904-f006:**
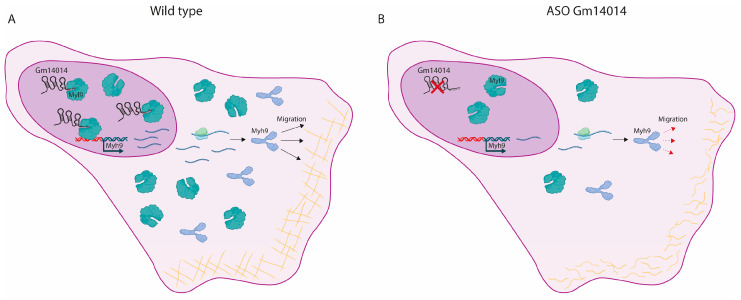
**Schematic representation of the working model of *Gm14014* during epicardial cell migration.** Panel (**A**) illustrates the role of *Gm14014* in homeostasis and in *Gm14014* silencing assays (Panel (**B**)). Note that *Gm14014* is bound to Myl9 in the nucleus, providing insight into its role in cell migration by interacting with Myh9 in the cytoplasm; however, if *Gm14014* is knocked down, no Myl9 nuclear binding occurs and thus it is completely translocated to the cytoplasm. Schemes were made with Biorender.

## Data Availability

The datasets generated during and/or analyzed during the current study are available from the corresponding author on reasonable request.

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
