# Peer review of "Mef2c- and Nkx2-5-Divergent Transcriptional Regulation of Chick WT1_76127 and Mouse Gm14014 lncRNAs and Their Implication in Epicardial Cell Migration"

_ijms, 2024, doi:10.3390/ijms252312904_

Round 1
Reviewer 1 Report
Comments and Suggestions for Authors
I reviewed the manuscript titled "Mef2c and Nkx2-5 Divergent Transcriptional Regulation of Chick WT1_76127 and Mouse Gm14014 lncRNAs and their Implication in Epicardial Cell Migration" by Caño-Carrillo et al., which dives into how two transcription factors, Mef2c and Nkx2-5, differentially regulate two lncRNAs—WT1_76127 in chickens and Gm14014 in mice. These lncRNAs play crucial roles in epicardial development and cell migration. I found it fascinating how the authors highlight the specific expression patterns of these lncRNAs during different developmental stages and within subcellular compartments, which suggests they may be closely involved in cytoskeletal regulation and cellular movement.
Through genetic inhibition techniques and migration assays, the authors demonstrate that Gm14014 regulation directly impacts epicardial cell migration in mice. The paper also points out evolutionary divergences in transcriptional regulation across species, which emphasizes the complexity of lncRNA roles in cardiac development. These insights could be quite valuable in the context of future therapies targeting cardiac repair and cell migration in heart disease. The study overall is well-conducted and compelling, though I do have some suggestions to enhance its depth and clarity.
Major comments
# I noticed that each figure could benefit from a general description to improve clarity. It would also be helpful to specify which software was used to create illustrations (biorender?), especially for Figures 1C and 6. Clear figure descriptions and information on software make it easier for readers to follow and replicate the study's findingss.
“The manuscript mentions the use of RT-qPCR, Western blot, and siRNA inhibition techniques. However, adding details on the positive and negative controls for these inhibition methods (e.g., siRNA, ASO) would strengthen the results by ensuring specificity and reliability. I recommend highlighting this in the methods section.
#While the subcellular isolation techniques are explained, including more specific details on the nuclear and cytoplasmic purification procedures would add clarity and ensure that readers understand the purity and integrity of these samples.
# For Figs. 1E-F and 3C-D, which address transcriptional regulation in different cells, the data suggests distinct roles for Mef2c and Nkx2-5. Adding a summary figure that visually represents the specific roles of each transcription factor in the nucleus and cytoplasm would improve the overall comprehension of this complex data.
# Figure 4: This figure focuses on epicardial cell migration assays with Gm14014 ASO treatment. Including additional microscopy images at different time points (e.g., 6h, 12h, 24h) could provide a visual narrative of cell migration progression rather than relying solely on final quantitative data.
# For the immunocytochemistry images showing cytoskeletal proteins (e.g., Myh9 and Myl9) in treated versus control cells, I feel that comparing fluorescence intensity in these images would allow readers to quantify the expression differences between treated and control cells. This could make the findings on protein expression more tangible.
#Fig. 4F appears quite small, making it difficult to distinguish individual lanes. Increasing its size could help readers better visualize the data presented.
#While the discussion is well-organized, it would benefit from a direct comparison of the findings with similar studies on lncRNAs in epicardial development in other models. Contextualizing these results with recent studies on lncRNAs in cardiac development and migration would provide a broader perspective and enhance the study's relevance..
# Regarding limitations, I suggest that the authors delve further into the observed differences in transcriptional regulation between chickens and mice. Discussing how these interspecies differences might impact the interpretation of the results in other systems would add valuable context.
# The authors propose that Gm14014 regulation may have a specific role in epicardial cell migration in mice. It would also be helpful if they suggested future research avenues, such as exploring whether inhibiting these lncRNAs impacts cardiac development or injury response in other animal models. Such suggestions would underline the translational potential of these findings.
Minor Issues:
#There are a few minor typos throughout the manuscript (e.g., "thymosin 4" should be consistent, and lowercase "p" in the abstract should be corrected).
Author Response
First of all, we would like to thanks the reviewer for his/her positive assessment of our study. We greatly appreciate his/her kind words on our work.
Major comments
# I noticed that each figure could benefit from a general description to improve clarity. It would also be helpful to specify which software was used to create illustrations (biorender?), especially for Figures 1C and 6. Clear figure descriptions and information on software make it easier for readers to follow and replicate the study's findingss.
We would like to thank the reviewer to highlight the attention on this point and thus following his/her recommendation, we have implemented a general description on each figure legend and we have provided more detailed information regarding the software used to create the illustrations.
“The manuscript mentions the use of RT-qPCR, Western blot, and siRNA inhibition techniques. However, adding details on the positive and negative controls for these inhibition methods (e.g., siRNA, ASO) would strengthen the results by ensuring specificity and reliability. I recommend highlighting this in the methods section.
Following the recommendation of the reviewer, we have added more detailed information regarding the positive and negative controls in the methods section of all of the experimental approaches used in this study that were missing this information, since in several cases, it was already available.
#While the subcellular isolation techniques are explained, including more specific details on the nuclear and cytoplasmic purification procedures would add clarity and ensure that readers understand the purity and integrity of these samples.
Following the recommendation of the reviewer, we have added more detailed information regarding the purification procedures used during subcellular isolation, in order to enhance the understanding of the purity and integrity of the samples in the revised version of the manuscript (methods section)
# For Figs. 1E-F and 3C-D, which address transcriptional regulation in different cells, the data suggests distinct roles for Mef2c and Nkx2-5. Adding a summary figure that visually represents the specific roles of each transcription factor in the nucleus and cytoplasm would improve the overall comprehension of this complex data.
As suggested by the reviewer, a summary figure has been added to improve the understanding of the transcriptional regulation of these lncRNAs in both Figures 1 and 3, respectively.
# Figure 4: This figure focuses on epicardial cell migration assays with Gm14014 ASO treatment. Including additional microscopy images at different time points (e.g., 6h, 12h, 24h) could provide a visual narrative of cell migration progression rather than relying solely on final quantitative data.
As suggested by the reviewer, additional microscopical images at different time points have been added on Figure 4 as well as in the Supplementary data, in the revised version of the manuscript.
# For the immunocytochemistry images showing cytoskeletal proteins (e.g., Myh9 and Myl9) in treated versus control cells, I feel that comparing fluorescence intensity in these images would allow readers to quantify the expression differences between treated and control cells. This could make the findings on protein expression more tangible.
We agree with the reviewer, and indeed these information is currently available in Figure 4, panel E. The graphical bars represent indeed the mean fluorescence intensity of each corresponding experimental condition imaged in the control and ASO experiments at 6h and 24h after administration, respectively.
#Fig. 4F appears quite small, making it difficult to distinguish individual lanes. Increasing its size could help readers better visualize the data presented.
Following the recommendation of the reviewer, we have increased the size of the images corresponding to Figure 4F in the revised version of the manuscript.
#While the discussion is well-organized, it would benefit from a direct comparison of the findings with similar studies on lncRNAs in epicardial development in other models. Contextualizing these results with recent studies on lncRNAs in cardiac development and migration would provide a broader perspective and enhance the study's relevance..
As suggested by the reviewer, we have edited the discussion to include the contextualization of our results with recent studies on lncRNAs in cardiac development and migration in other experimental models. However, to the best of our knowledge, no information is available about the role of lncRNAs in epicardial development and thus, we have reflected as such in the revised version of the manuscript.
# Regarding limitations, I suggest that the authors delve further into the observed differences in transcriptional regulation between chickens and mice. Discussing how these interspecies differences might impact the interpretation of the results in other systems would add valuable context.
Following the recommendation of the reviewer, we have done our best to increase the discussion of the interspecies differences in the transcriptional regulation of lncRNAs, since there is a very limited number of studying transcriptional regulation of lncRNAs at large and none, to the best of our knowledge, reporting interspecies differences. Thus, our study will be the first one reporting such differences.
# The authors propose that Gm14014 regulation may have a specific role in epicardial cell migration in mice. It would also be helpful if they suggested future research avenues, such as exploring whether inhibiting these lncRNAs impacts cardiac development or injury response in other animal models. Such suggestions would underline the translational potential of these findings.
Following the recommendation of the reviewer, we have added in the discussion some insights about the future research avenues, which are indeed already been initiated in our laboratory, as look at the role of this lncRNA in distinct model of cardiac injury and pathology and developing systemic gene edited Gm14014 mouse mutants. These experimental approaches will certain enhance the translational potential of our findings.
Minor Issues:
#There are a few minor typos throughout the manuscript (e.g., "thymosin 4" should be consistent, and lowercase "p" in the abstract should be corrected).
Following the recommendation of the reviewer, we have corrected these minor typos and we have carefully revised the entire manuscript.
Reviewer 2 Report
Comments and Suggestions for Authors
The article addresses an important and novel area in cardiovascular research, exploring the role of lncRNAs in epicardial cell migration and its implications for heart development and regeneration. It provides detailed experimental data and compelling evidence for species-specific roles of homologous lncRNAs. The methodology is robust, and the findings are significant for advancing our understanding of molecular mechanisms in cardiac biology. However, several aspects can be improved to enhance clarity, accessibility, and impact.
- Terms like "cytoskeletal remodeling" and "cell migration" are used interchangeably without clear definitions. Clarify distinctions and maintain consistency throughout the text.
- While statistical analyses are mentioned, details about statistical tests and confidence intervals are sparse. Provide explicit statistical methods and parameters in all figure legends
- The manuscript does not adequately compare its findings with existing literature on lncRNAs in other developmental or regenerative contexts. Including this would strengthen the discussion
- The species-specific differences between chicken and mouse models are interesting but require more in-depth discussion about their evolutionary and functional implications
- A thorough language edit is recommended to improve readability
Author Response
First of all, we would like to thanks the reviewer for his/her positive assessment of our study. We greatly appreciate his/her kind words on our work.
-Terms like "cytoskeletal remodeling" and "cell migration" are used interchangeably without clear definitions. Clarify distinctions and maintain consistency throughout the text.
We fully agree with the reviewer that the terms "cytoskeletal remodeling" and "cell migration" should not be used as interchangeable synonyms Following his/her recommendation we have carefully checked the entire manuscript to delete such misusage. We have indeed clarify the distinction between them and provide consistency throughout the text in their usage, whenever required.
-While statistical analyses are mentioned, details about statistical tests and confidence intervals are sparse. Provide explicit statistical methods and parameters in all figure legends
Following the recommendation of the reviewer, we have added more detailed information and the statistical tests used and their confidence interval in both, the material and methods section and within the figure legends, as requested by the reviewer, in the revised version of the manuscript.
-The manuscript does not adequately compare its findings with existing literature on lncRNAs in other developmental or regenerative contexts. Including this would strengthen the discussion
As suggested by the reviewer, we have edited the discussion to include the contextualization of our results with recent studies on lncRNAs in cardiac development, regeneration and migration. However, to the best of our knowledge, no information is available about the role of lncRNAs in epicardial development and thus, we have reflected as such in the revised version of the manuscript.
-The species-specific differences between chicken and mouse models are interesting but require more in-depth discussion about their evolutionary and functional implications
Following the recommendation of the reviewer, we have done our best to increase the discussion of the interspecies differences in the transcriptional regulation of lncRNAs, since there is a very limited number of studying transcriptional regulation of lncRNAs at large and none, to the best of our knowledge, reporting interspecies differences. Thus, our study will be the first one reporting such differences.
-A thorough language edit is recommended to improve readability
Following the recommendation of the reviewer, we carefully revised the entire manuscript with the aim of improving readability.
Round 2
Reviewer 1 Report
Comments and Suggestions for Authors
I am satisfied with the authors' responses and believe they have made significant changes to their manuscript. Suitable for publication
Reviewer 2 Report
Comments and Suggestions for Authors
The authors took into account the recommendations of the reviewers and made improvements to the manuscript, thus making it suitable for publication.